# Effect of Same-Sex Marriage Referendums on the Suicidal Ideation Rate among Nonheterosexual People in Taiwan

**DOI:** 10.3390/ijerph16183456

**Published:** 2019-09-17

**Authors:** I-Hsuan Lin, Nai-Ying Ko, Yu-Te Huang, Mu-Hong Chen, Wei-Hsin Lu, Cheng-Fang Yen

**Affiliations:** 1Department of Psychiatry, Yuan’s General Hospital, Kaohsiung 80249, Taiwan; ihreneelin@gmail.com; 2Departments of Nursing, College of Medicine, National Cheng Kung University and Hospital, Tainan 70101, Taiwan; nyko@mail.ncku.edu.tw; 3Center of Infection Control, National Cheng Kung University Hospital, Tainan 70101, Taiwan; 4Department of Social Work and Social Administration, The University of Hong Kong, Hong Kong RM543, Hong Kong; Yuhuang@hku.hk; 5Department of Psychiatry, Taipei Veterans General Hospital, Taipei 11217, Taiwan; kremer7119@gmail.com; 6Division of Psychiatry, School of Medicine, National Yang-Ming University, Taipei 11221, Taiwan; 7Department of Psychiatry, Ditmanson Medical Foundation Chia-Yi Christian Hospital, Chia-Yi City 60002, Taiwan; 8Department of Senior Citizen Service Management, Chia Nan University of Pharmacy and Science, Tainan 60002, Taiwan; 9Department of Psychiatry, School of Medicine, College of Medicine, Kaohsiung Medical University, Kaohsiung 80708, Taiwan; 10Department of Psychiatry, Kaohsiung Medical University Hospital, Kaohsiung 80708, Taiwan

**Keywords:** age, gender, same-sex marriage, sexual orientation, suicidality

## Abstract

Taiwan held voter-initiated referendums to determine same-sex marriage legalization on 24 November 2018. This study aims to compare suicidal ideation rates in heterosexual and nonheterosexual participants of a first-wave survey (Wave 1, 23 months before the same-sex marriage referendums) and a second-wave survey (Wave 2, one week after the same-sex marriage referendums) in Taiwan and to examine the influence of gender, age, and sexual orientation on the change in suicidal ideation rates in nonheterosexual participants. In total, 3286 participants in Wave 1 and 1370 participants in Wave 2 were recruited through a Facebook advertisement. Each participant completed an online questionnaire assessing suicidal ideation. The proportions of heterosexual and nonheterosexual participants with suicidal ideation were compared between the Wave 1 and Wave 2 surveys. Suicidal ideation rates between participants in the Wave 1 and Wave 2 surveys were further compared by stratifying nonheterosexual participants according to gender, age, and sexual orientation. Nonheterosexual participants in the Wave 2 survey had a higher suicidal ideation rate than those in the Wave 1 survey, whereas no difference was observed in suicidal ideation rates between heterosexual participants in Wave 2 and Wave 1. Nonheterosexual participants who were female, younger, gay, lesbian, and bisexual in Wave 2 had a higher suicidal ideation rate than those in Wave 1. The suicidal ideation rate significantly increased in nonheterosexual participants experiencing the same-sex marriage referendums in Taiwan. Whether civil rights of sexual minority individuals can be determined through referendums should be evaluated.

## 1. Introduction

### 1.1. Suicide in Sexual Minorities

Suicide is a critical health issue among sexual minority individuals. A meta-analysis found that sexual minority youth reported significantly higher rates of suicidality than did their heterosexual counterparts [1]. A meta-analysis pooling 19 studies found that the prevalence of lifetime suicidal ideation in men who have sex with men was 34.97%, which is far higher than that in the general population [2]. Another meta-analysis pooling 30 studies found that sexual minority adults had nearly three- to five-times higher risks of suicidal attempts than did heterosexual individuals [3]. Research has found that most suicidal attempts are preceded by suicidal ideation [4,5]. Therefore, suicidal ideation warrants careful evaluation and intervention to prevent eventual suicide completion.

### 1.2. Same-Sex Marriage Bans: A Structural-Level Discrimination toward a Sexual Minority

According to the ecological systems theory [6], suicidality may result from complex interactions between sexual minority individuals and their environments. One of the individual–environmental interacting factors that may increase the suicidal risk of sexual minority individuals is stigma based on sexual orientation [7]. According to minority stress theory [8], socially-stigmatized individuals may experience chronic stress due to their minority statuses and consequently develop mental health problems. Sexual minority individuals may internalize sexuality-related oppression and experience stress caused by hiding and managing a socially-stigmatized identity, both of which further compromise their mental health [9]. In addition to perceived discrimination [10], the expectation of being discriminated against by others [11], internalized stigma [12], and structural stigma [13] has been identified as a contributor to mental health problems in sexual minority individuals. Same-sex marriage bans are one type of structural-level discrimination that differentially targets sexual minority individuals due to social exclusion and compromises their mental health [14,15]. Lesbian, gay, and bisexual (LGB), but not heterosexual, individuals living in states in the United States (U.S.) that passed constitutional amendments banning same-sex marriage experienced significant increases in mood disorder, generalized anxiety disorder, alcohol use disorder, and psychiatric comorbidity; the increase in psychiatric disorders was not found among LGB individuals living in states without these constitutional amendments [14]. LGB individuals reported that constitutional amendments banning same-sex marriage make them feel indignant about discrimination, as well as fearful, anxious, and hopeless about protecting their relationships and families [15]. These results demonstrated the harmful discriminating effects of passing same-sex marriage bans on the mental health of sexual minority individuals.

### 1.3. Same-Sex Marriage Campaign and Referendums in Taiwan

Sexual minority rights campaigners in Taiwan have strived for same-sex marriage legalization since the end of the 1980s. However, people in Taiwan traditionally regard homosexuality as a challenge to the family obligations mandated in Confucianism, and in particular, they require their offspring to continue the family bloodline. Moreover, the Civil Code’s stipulation “an agreement to marry shall be made by the male and the female parties in their own concord” renders same-sex marriage difficult to legalize [16]. In the past two decades, overall, an attitude of social tolerance toward homosexuality has become widespread in Taiwan, which is mainly accounted for by improvement in education and liberal values related to gender roles [17]. The 2012 Taiwan Social Change Survey showed that for the first time, supporters of same-sex marriage outnumber those who oppose it [18]. The most encouraging progress of same-sex marriage legalization in Taiwan is that in May 2017, the Council of Grand Justices announced that the current Civil Code that barred same-sex marriage is a violation of the human right to equality and is unconstitutional, and the council directed that same-sex marriage should be legalized within two years. With such progress, sexual minority rights campaigners in Taiwan rejoiced at the prospect of same-sex marriage.

However, the progress of same-sex marriage has drawn substantial opposition, mainly from Christian groups, in Taiwan. In response to the ruling of the Council of Grand Justices on same-sex marriage, the group against same-sex marriage drafted two referendums to argue that legal reform should be made outside changes to the Civil Code, including Case No. 10: “Do you agree that marriage defined in the Civil Code should be restricted to the union between one man and one woman?" and Case No. 12: "Do you agree to the protection of the rights of same-sex couples in co-habitation on a permanent basis in ways other than changing of the Civil Code?”. By contrast, the group lobbying for marriage equality drafted a referendum (Case No. 14: "Do you agree to the protection of same-sex marital rights with marriage as defined in the Civil Code?") to argue that separate legislation amounts to a form of discrimination. The results of the vote on 24 November 2018, indicated that Case No. 10 and Case No. 12 received overwhelming support, with 70.12% and 57.60% of voters in favor, respectively. By contrast, only 30.27% of voters supported Case No. 14. The results of voting suggested that the two referendums drafted by the group against same-sex marriage received considerably stronger support than the one by the group supporting marriage equality.

Research has shown that Asian countries exhibit considerably less tolerance for homosexuality than do European and North American countries [19]. Taiwan is the first Asian country to deliberate on same-sex marriage legalization through voter-initiated referendums. Nevertheless, the results of the referendums on 24 November 2018, definitely discouraged sexual minority individuals and minority rights campaigners. Given that voter-initiated referendums occur with some regularity and affect numerous minority groups [20], the effects of the same-sex marriage ban referendums on the mental health of sexual minority individuals in Taiwan warrants further study. The result of such a study may provide empirical evidence to understand the impacts of voter-initiated referendums on mental health in minority groups whose rights are restricted or rejected, as well as to inspect whether civil rights of any individual can be determined through referendums.

### 1.4. Aims and Hypotheses of this Study

The Investigation on the Attitude Toward Same-Sex Marriage in Taiwan is a two-wave online survey of people’s attitude toward same-sex marriage and the mental health status of sexual minority individuals in Taiwan. The first wave (Wave 1) was conducted from 1–31 January 2017, 23 months before the same-sex marriage referendums. The second wave (Wave 2) was conducted from 1–31 December 2018, one week after the same-sex marriage referendums. The two aims and corresponding hypotheses of the present study are described below.

### 1.5. Aim I: To Compare the Suicidal Ideation Rate in Nonheterosexual and Heterosexual Participants between the Wave 1 and Wave 2 Surveys

The negative results of the same-sex marriage referendums directly discriminated and devaluated nonheterosexual people. Moreover, the groups opposing same-sex marriage spent a large amount of money to malign the image of sexual minority individuals through propaganda on mass media and social media. Research has found that greater exposure to same-sex marriage campaign advertisements is associated with high stress in sexual minority individuals, and negative advertisements evoke the feeling of sadness among them [20]. The same-sex marriage referendum process and result may also cause the exposure of sexual minority individuals to hostile interactions with neighbors, colleagues, and family members [21]. By contrast, heterosexual individuals are spared from developing mental health problems related to same-sex marriage bans [14]. Therefore, in the present study, we hypothesized that the suicidal ideation rate in nonheterosexual participants increased from the Wave 1 to Wave 2 surveys, whereas no significant difference was observed in suicidal ideation rates among heterosexual participants between the Wave 1 and Wave 2 surveys.

### 1.6. Aim II: To Examine the Effects of Gender, Age, and Sexual Orientation on Differences in Suicidal Ideation Rates among Nonheterosexual Participants between the Wave 1 and Wave 2 Surveys

Research has demonstrated the positive effects of formal same-sex relationships on psychological well-being in younger, but not in older, lesbians and gay men [22]. Research has also found that civil union legalization is the most beneficial for racial or ethnic minority women and women with lower levels of education [23]. Whether same-sex marriage referendums exert various psychological effects on nonheterosexual individuals with various genders, ages, and sexual orientations warrants further study. We hypothesized that differences existed in suicidal ideation rates among nonheterosexual participants of various genders, ages, and sexual orientations.

## 2. Methods

### 2.1. Participants

The method of recruiting participants is described elsewhere [24]. In brief, participants aged at least 20 years were recruited into the two-wave online survey through a Facebook advertisement. The Facebook advertisement included a headline, main text, pop-up banner, and weblink to the study questionnaire website. The advertisement appeared in the News Feed of Facebook, which is a streaming list of updates from the user’s connections and advertisers. News feed advertisements are more effective in terms of recruitment metrics for studies [25]. We targeted the advertisement to Facebook users by location (Taiwan) and language (Chinese). The de-duplication protocol used in the present study to identify multiple submissions and preserve data integrity included cross-validation of the eligibility of key variables and examination of discrepancies in key data, as well as checking for unusually fast completion time (<10 minutes) [26]. Moreover, each Internet Protocol address could be registered to complete the online questionnaire once only.

Participants were not given any incentives for participation. All subjects gave their informed consent for inclusion before they participated in the study. The study was conducted in accordance with the Declaration of Helsinki, and the protocol was approved by the Ethics Committee of Kaohsiung Medical University Hospital (KMUHIRB-EXEMPT(II)-20160065). The study design involved respondents’ online response to the recruitment advertisement and questionnaire anonymously, which allowed the respondents to decide freely whether to join or not, and their personal information was kept secure. Owing to the anonymity of participants, we could not determine how many participants responded to both surveys. Therefore, the data of the two waves of the survey was analyzed independently. The IRB thus agreed that this study did not require obtaining informed consent from the respondents.

### 2.2. Measures

#### 2.2.1. Suicidal Ideation

We used the question “Do you have any suicide ideation?” on the Revised 5-item Brief Symptom Rating Scale to inquire participants’ suicidal ideation during the past week. Participants were asked to rate the severity of suicidal ideation on a 5-point scale: 0, not at all; 1, a little bit; 2, moderately; 3, quite a bit; and 4, extremely [27]. Participants who rated ≥2 on the item were classified as having significant suicidal ideation.

#### 2.2.2. Demographic Variables

Data on participants’ gender (female, male, and transgender), age, and sexual orientation (heterosexual, bisexual, homosexual, pansexual, asexual, and unsure) were collected. According to sexual orientation, participants were classified into heterosexual and nonheterosexual (including bisexual, homosexual, and others) groups. According to age, participants were classified into the age groups of 20–29, 30–39, and ≥40 years.

### 2.3. Procedure and Statistical Analysis

The proportions of gender, age, and suicidal ideation were compared between the Wave 1 and Wave 2 surveys in heterosexual and nonheterosexual groups by using the χ^2^ test. Because of multiple comparisons, a *p*-value of <018 (0.05/3) was considered statistically significant for all tests. The proportions of nonheterosexual participants with suicidal ideation were compared between Wave 1 and Wave 2 surveys in various gender (female, male, and transgender), age (20–29, 30–39, and ≥40 years), and sexual orientation groups (homosexual, bisexual, and others) by using the χ^2^ test. Because of multiple comparisons, a *p*-value of <006 (0.05/9) was considered statistically significant for all tests.

## 3. Results

A total of 3423 and 1395 Facebook users completed the online questionnaire in Wave 1 and Wave 2, respectively. Among them, 137 and 25 were excluded from analysis because they were underage (<20 years) or had an erroneous value for age (>100 years) in Wave 1 and Wave 2, respectively. The final data of 3286 participants (1456 heterosexual and 1830 nonheterosexual individuals) in Wave 1 and 1370 participants (540 heterosexual and 830 nonheterosexual individuals) in Wave 2 was analyzed. Table 1 shows the results of a comparison of demographic characteristics between participants in the Wave 1 and Wave 2 surveys. In nonheterosexual groups, higher numbers of transgender individuals were found in the Wave 2 (3.9%) than in Wave 1 (1.9%) survey (χ^2^ = 9.488, *p* = 0.009). Higher numbers of heterosexual participants aged 20–29 years (44.0% vs. 29.1%) and lower numbers of heterosexual participants aged ≥40 years (19.2% vs. 35.2%) were found in the Wave 1 survey than in the Wave 2 survey (χ^2^ = 64.554, *p* < 0.001).

### 3.1. Change in Suicidal Ideation Rates between Heterosexual and Nonheterosexual Participants

Table 1 also shows the results of a comparison of suicidal ideation rates between participants in the Wave 1 and Wave 2 surveys. Nonheterosexual participants in the Wave 2 survey (24.6%) had a higher suicidal ideation rate than nonheterosexual participants in the Wave 1 survey (15.4%) (χ^2^ = 32.145, *p* < 001), whereas no difference was observed in suicidal ideation rates between heterosexual participants in Wave 1 (6.3%) and Wave 2 (5.2%) surveys (χ^2^ = 877, *p* = 349).

### 3.2. Changes in Suicidal Ideation Rates in Nonheterosexual Participants of Various Genders, Ages, and Sexual Orientations

Table 2 shows the results of a comparison of suicidal ideation rates in nonheterosexual participants of various genders, ages, and sexual orientations between the Wave 1 and Wave 2 surveys. Nonheterosexual women exhibited a significant increase in suicidal ideation rates from the Wave 1 to Wave 2 surveys (14.0% vs. 36.4%, χ^2^ = 26.125, *p* < 0.001). Suicidal ideation rates in nonheterosexual men tended to increase from the Wave 1 to Wave 2 surveys (16.8% vs. 23.3%, χ^2^ = 7.371, *p* = 0.007), but the difference was not statistically significant. No significant increase in suicidal ideation rates was detected in nonheterosexual transgender individuals from the Wave 1 to Wave 2 surveys (17.6% vs. 28.1%, χ^2^ = 1.031, *p* = 0.310).

Nonheterosexual participants aged 20–29 years (17.0% vs. 27.3%, χ^2^ = 21.642, *p* < 0.001) and aged 30–39 years (13.7% vs. 22.6%, χ^2^ = 10.837, *p* = 0.001) exhibited higher suicidal ideation rates in the Wave 2 survey than in the Wave 1 survey. No difference was observed in the suicidal ideation rate in older nonheterosexual participants (aged ≥40 years) between Wave 1 and Wave 2 (10.4% vs. 15.2%, χ^2^ = 1.092, *p* = 0.296).

Gay and lesbian (16.6% vs. 26.4%, χ^2^ = 21.838, *p* < 0.001) and bisexual participants (11.1% vs. 23.2%, χ^2^ = 15.408, *p* < 0.001) exhibited higher suicidal ideation rates in the Wave 2 survey than in the Wave 1 survey. No difference was observed in the suicidal ideation rate in the participants with pansexual, asexual, and unsure sexual orientations between Wave 1 and Wave 2 (17.1% vs. 18.1%, χ^2^ = 0.052, *p* = 0.820).

## 4. Discussion

The results of the present study revealed that nonheterosexual participants in the Wave 2 survey had a higher suicidal ideation rate than those in the Wave 1 survey, whereas no difference was found in suicidal ideation rates in heterosexual participants between Wave 2 and Wave 1. Nonheterosexual participants who were female, younger (aged 20–39 years), gay, lesbian, and bisexual in Wave 2 had a higher suicidal ideation rate than those in Wave 1.

### 4.1. Suicidal Ideation in LGB Participants Experiencing Same-Sex Marriage Referendums

The present study found that the suicidal ideation rate in nonheterosexual participants significantly increased from Wave 1 (conducted 23 months before the same-sex marriage referendums) to Wave 2 (conducted one week after the same-sex marriage referendums), whereas the suicidal ideation rate did not significantly change in heterosexual participants. The same-sex marriage referendums might specifically influence the suicidal ideation rate among sexual minority individuals in Taiwan in two ways: the campaigns against same-same marriage before the referendums and the negative results of the referendums. First, the groups opposing same-sex marriage in Taiwan spread a considerable amount of incorrect information and rumors to malign same-sex marriage and sexual minority individuals through social media and public media, as they proposed the referendums against same-sex marriage. For example, they claimed that the legalization of same-sex marriage would lead to the widespread outbreak of the contagion of human immunodeficiency virus infection, depopulation in Taiwan, and the deterioration of traditional family values. These misleading portrayals and negative stereotypes spread in the media demoralized sexual minority individuals and directly disturbed their emotional regulation. Research in the U.S. found that exposure to negative same-sex marriage campaign advertisements evoked the feeling of sadness among lesbian, gay, bisexual, and/or transgender individuals [20]. Research in Australia found that more frequent exposure to negative media messages about same-sex marriage was associated with greater psychological distress [28]. Moreover, LGB individuals might internalize distorted images and point of views into their self-appraisals and feel ashamed of their LGB identity [29].

Second, although the 2012 Taiwan Social Change Survey found that supporters of same-sex marriage outnumbered those opposing it [18], the large amount of false information broadcast by anti-LGB campaigners certainly influenced the values of people in Taiwan to a certain extent. Research has shown that public campaigns debating anti-gay policies, such as same-sex marriage, may foster a negative social climate for sexual minority individuals [30]. After the proposal of referendums in Taiwan, sexual minority individuals had to interact with neighbors, colleagues, and family members who adopted the viewpoints broadcast by the groups opposing same-sex marriage for half a year. According to the social identity threat theories of stigma [31], cues from the social environment that are appraised as potentially harmful to one’s stigmatized social identity engender a threat, which in turn creates involuntary stress responses. Stigma-related stress deteriorates victims’ emotion dysregulation and cognitive processes and further confers the risk of psychopathology [32]. A previous study had a similar result that LGB people reported comparatively worse life satisfaction, mental health, and overall health in constituencies with higher rates of voters saying “no” to the same-sex plebiscite [33].

Third, the result that a certain group of voters favored that sexual minority individuals only have the right to cohabit but not marry according to the Civil Code definitely discriminated between sexual minority individuals and heterosexual individuals. Creating laws ruling that sexual minority individuals do not have the same rights as heterosexual individuals reinforced the marginalized and socially-devalued statuses of sexual minority individuals [21,30]. The European Social Survey determined that sexuality-based discrimination has significant negative effects on the self-related health and subjective well-being of victims [34]. The results of the present study support that referendums on the civil rights of sexual minority individuals represent a source of stress for this sexual minority and may have significant negative effects on the mental health of sexual minority individuals and increases their suicide risk. Whether the civil rights of sexual minority individuals can be determined through voter-initiated referendums should be comprehensively evaluated. Mental health professionals must develop prevention and intervention strategies for suicide risk in LGB individuals experiencing referendums that decide their civil rights.

### 4.2. Gender Differences in the Change in Suicidal Ideation Rates

The present study found that nonheterosexual women had significantly exhibited higher suicidal ideation rates in the Wave 2 survey than in the Wave 1 survey (*p* < 0.001), whereas suicidal ideation rates in nonheterosexual men tended to increase, but not significantly (*p* = 0.007). This gender difference might be partially attributed to the double stigma that many lesbians experience as both lesbians and women [35]. Taiwanese society considers women subordinate to men. In the past decade, women’s reproductive health, empowerment, and labor market have improved significantly [36]. However, the gender gap in social status remains nonuniform. For example, the gender pay gap still exists in Taiwan, with women earning 85.4% of the average hourly income of men [36]. Moreover, a longitudinal study in Australia found that nonheterosexual women were more disadvantaged in health and wellbeing than nonheterosexual men [37]. As a structural stigma, the result of the referendums may interact with individual disadvantages, including sexual minority and underprivileged gender, which may cause lesbian individuals to become vulnerable to the frustration caused by failure in changing the Civil Code for same-sex marriage legalization. The result indicated the importance of considering gender differences in psychological reactions to major events related to sexual minority rights.

In the present study, suicidal ideation rates did not significantly increase in nonheterosexual transgender individuals from the Wave 1 to Wave 2 surveys. The small number of nonheterosexual transgender participants in the present study limited the possibility of drawing a conclusion on the effect of the same-sex marriage referendums on the mental health of nonheterosexual transgender individuals. Subgroups of nonheterosexual individuals may have had various experiences during the period of the same-sex marriage referendums. The questions of whether nonheterosexual transgender individuals feel marginalized in the debates and whether they consequently feel uninvolved in the same-sex marriage movement warrant further study.

### 4.3. Age Differences in the Change in Suicidal Ideation Rates

The present study found that nonheterosexual participants aged 20–29 years exhibited the most significant increase in the suicidal ideation rate, followed by those aged 30–39 years, from the Wave 1 to Wave 2 surveys, whereas no difference was observed in the suicidal ideation rate in those aged ≥40 years (*p* = 0.296). Research has demonstrated that young people have a more tolerant attitude toward homosexuality than older people in Taiwan [17]. Younger nonheterosexual participants may have an overly optimistic expectation of legalizing same-sex marriage based on the atmosphere they perceived from their peers, whereas older nonheterosexual participants may have a pessimistic expectation based on the social stigma prevailing in Taiwan for a long time. Various expectations may result in various levels of shock and disappointment in response to the results of the referendums and further caused the difference in changes in suicidal ideation rates in various age groups of nonheterosexual participants. Moreover, nonheterosexual participants in early adulthood may still strive to establish their sexual identity and self-worth. The result of the referendums may disappoint their establishment of sexual identity and compromise their psychological well-being. The result indicated the importance of considering age when developing prevention and intervention programs for suicidality in nonheterosexual individuals experiencing the legalization of policies hostile to any sexual minority.

### 4.4. Sexual Orientation Differences in the Change in Suicidal Ideation Rates

The present study found that gay, lesbian, and bisexual, but not pansexual, asexual, and unsure, participants had increased suicidal ideation rates from the Wave 1 to Wave 2 surveys. No epidemiological study has examined the proportions of pansexual, asexual, and unsure individuals in Taiwan. In total, 7.3% and 7.7% of participants in the Wave 1 and Wave 2 surveys, respectively, labeled their sexual orientation as pansexual, asexual, or unsure; these proportions were lower than those of individuals who labeled themselves as homosexual (35.5% in Wave 1 and 38.8% in Wave 2) and bisexual (12.9% in Wave 1 and 14.2% in Wave 2). Individuals whose sexual orientations are pansexual, asexual, or unsure are minor groups in the sexual minority. Their experiences in the debates of same-sex marriage require additional studies to deepen the understanding of various sexual minority groups.

### 4.5. Limitations

The present study has some limitations. First, although recruiting participants through Facebook can deliver large numbers of participants quickly, cheaply, and with minimal effort compared with mail and phone recruitment, access to Facebook is not universal to people of all ages. A 2018 analysis found that 68.4% of active Facebook users in Taiwan were aged between 18 and 44 [38]. Moreover, people are not equally motivated to use Facebook [39]. Although the female:male ratio of Facebook uses in Taiwan is about 1:1 [38], no data shows the distribution of sexual orientation in Facebook users in Taiwan. Therefore, whether young lesbians are more likely to participate in Facebook as a way of creating a social community and more likely to express suicidal ideation warrants further study.

Second, the distributions of heterosexual and nonheterosexual participants in the current study were not congruent with those in the general population. The number of female heterosexual respondents was higher than that of male heterosexual respondents in both waves of the survey.

Third, the Wave 2 survey was conducted one week after the referendums. The nonheterosexual participants might be in a state of great anger and disappointment. Studies with relatively long follow-up periods are needed to examine the longer term change in suicidal ideation.

Fourth, participants’ suicidal ideation might develop in various biological, cognitive, and emotional contexts. The present study did not clarify the mechanisms for the increased suicidal ideation rate in nonheterosexual participants. The results of previous studies may provide possible explanations for the mechanisms through which voter referendums affect the mental health of sexual minority individuals in the U.S. [20,29,40]. However, whether these mechanisms proposed based on the U.S. sociocultural background can well explain the increased suicidal ideation rate in nonheterosexual participants in Taiwan warrants further study.

## 5. Conclusions

The suicidal ideation rate significantly increased in nonheterosexual individuals affected by the same-sex marriage referendums in Taiwan. Nonheterosexual participants who were female, younger, gay, lesbian, and bisexual were particularly vulnerable to the effects of the same-sex marriage referendums and had an increased suicidal ideation rate. The result indicated that the same-sex marriage ban referendums had a negative effect on the mental health of sexual minority individuals in Taiwan. The results also indicated the importance of considering gender, age, and sexual orientation differences in psychological reactions to major events related to sexual minorities. In addition to the inspection of whether civil rights of sexual minority individuals can be determined through referendums, factors that can protect sexual minority individuals from the hurt of structural stigma such as same-sex marriage bans warrant study. For example, research found that perceiving a greater immediate social network can buffer the effect of exposure to negative media messages about same-sex marriage on psychological distress [28]. Perceived poor social support also mediates a large portion of the effects of structural stigma on LGB outcomes [33].

## Figures and Tables

**Table 1 ijerph-16-03456-t001:** Comparison of demographic characteristics and suicidal ideation rates in heterosexual and nonheterosexual participants between the Wave 1 and Wave 2 surveys.

Variables	Heterosexual				Non-Heterosexual			
	Wave 1 (*n* = 1456) *n* (%)	Wave 2 (*n* = 540) *n* (%)	χ^2^	*p*	Wave 1 (*n* = 1830) *n* (%)	Wave 2 (*n* = 830) *n* (%)	χ^2^	*p*
Gender								
Female	1132 (77.8)	416 (77.0)	3.202	0.202	917 (50.1)	412 (49.6)	9.488	0.009
Male	311 (21.4)	123 (22.8)			879 (48.0)	386 (46.5)		
Transgender	13 (0.9)	1 (0.2)			34 (1.9)	32 (3.9)		
Age (years)								
20–29	640 (44.0)	157 (29.1)	64.554	<0.001	1075 (58.7)	472 (56.9)	2.207	0.332
30–39	536 (36.8)	193 (35.7)			611 (33.4)	279 (33.6)		
40 or older	280 (19.2)	190 (35.2)			144 (7.9)	79 (9.5)		
Suicidal ideation								
No	1380 (94.8)	506 (93.7)	0.877	0.349	1548 (84.6)	626 (75.4)	32.145	<0.001
Yes	76 (5.2)	34 (6.3)			282 (15.4)	204 (24.6)		

**Table 2 ijerph-16-03456-t002:** Comparison of suicidal ideation rates in nonheterosexual participants between the Wave 1 and Wave 2 surveys: gender, age, and sexual orientation effects.

Variables	Suicidal idea	χ^2^	*p*
Yes*n* (%)	No*n* (%)
Gender				
Female				
Wave 1 (*n* = 917)	128 (14.0)	789 (86.0)	26.125	<0.001
Wave 2 (*n* = 412)	105 (36.4)	307 (74.5)		
Male				
Wave 1 (*n* = 879)	148 (16.8)	731 (83.2)	7.371	0.007
Wave 2 (*n* = 386)	90 (23.3)	296 (76.7)		
Transgender				
Wave 1 (*n* = 34)	6 (17.6)	28 (82.4)	1.031	0.310
Wave 2 (*n* = 32)	9 (28.1)	23 (71.9)		
Age (years)				
20-29				
Wave 1 (*n* = 1075)	183 (17.0)	892 (83.0)	21.642	<0.001
Wave 2 (*n* = 472)	129 (27.3)	343 (72.7)		
30-39				
Wave 1 (*n* = 611)	84 (13.7)	527 (86.3)	10.837	0.001
Wave 2 (*n* = 279)	63 (22.6)	216 (77.4)		
40 or older				
Wave 1 (*n* = 144)	15 (10.4)	129 (89.6)	1.092	0.296
Wave 2 (*n* = 79)	12 (15.2)	67 (84.8)		
Sexual orientation				
Homosexual				
Wave 1 (*n* = 1166)	194 (16.6)	972 (83.4)	21.838	<0.001
Wave 2 (*n* = 531)	140 (26.4)	391 (73.6)		
Bisexual				
Wave 1 (*n* = 424)	47 (11.1)	377 (88.9)	15.408	<0.001
Wave 2 (*n* = 194)	45 (23.2)	149 (76.8)		
Others (pansexual, asexual and unsure)				
Wave 1 (*n* = 240)	41 (17.1)	199 (82.9)	0.052	0.820
Wave 2 (*n* = 105)	19 (18.1)	86 (81.9)

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
