# Peer review of "Effect of Same-Sex Marriage Referendums on the Suicidal Ideation Rate among Nonheterosexual People in Taiwan"

_ijerph, 2019, doi:10.3390/ijerph16183456_

Round 1

Reviewer 1 Report

This is excellent paper relevant to Taiwan and the rest of world.

Unfortunate that no population based studies have yet been conducted to determine how well social media, like Facebook, respondents represent the general population.

Author Response

Thank you for your comments. We revised the description regarding the limitation of recruiting participants in the present study as below: “First, although recruiting participants through Facebook can deliver large numbers of participants quickly, cheaply, and with minimal effort compared with mail and phone recruitment, access to Facebook is not universal to people of all age. A 2018 analysis found that 68.4% of active Facebook users in Taiwan were aged between 18 and 44 [38]. Moreover, people are not equally motivated to use Facebook [39]. Although the female: male ratio of Facebook uses in Taiwan is about 1:1 [38], no data shows the distribution of sexual orientation in Facebook users in Taiwan. Therefore, whether young lesbians are more likely to participate in Facebook as a way of creating a social community and more likely to express suicidal ideation warrants further study.” Please refer to line 343-351.

Reviewer 2 Report

There are some small issues with the use of language that need to be addressed. My biggest question is why LGB rather than LGBT or, LGBTQI? I am concerned that trans* is completely omitted because in this context it seems that people who identify as trans* would have a stake in this.

Author Response

Thank you for your suggestion. In the revised manuscript we replaced “LGB” by “sexual minority” in most of contents and kept using “LGB” if the cited studies focused on LGB population. Please to throughout the manuscript.

Reviewer 3 Report

Thank you for the opportunity to review Effect of same-sex marriage referendums on the suicidal ideation rate among nonheterosexual people in Taiwan. The paper addresses an important issue among gender and sexually diverse populations, and is creative in the way it approaches the problem. My concerns with the paper lie not so much with the design of the project but with the way it is presented. Most of these concerns can be readily addressed by the authors.

Firstly, while I believe the authors of the paper are well-intended, same-sex sexuality and marriage equality is presented as a problem in the paper. This is to some extent inevitable, since the purpose of the paper is to consider a problem, but there is little recognition of the resiliency of gender and sexually diverse communities in Taiwan. To what extent is the expression of suicidal ideation a proxy for expressions of sadness, anger, or even disappointment? I am not suggesting that expressions of suicidal intent not be taken seriously. However, in English, for example, the expression “I was so embarrassed I could have died!” is not an expression of a wish to die, but rather an expression of the extent of embarrassment. Since the Wave 2 study was carried out only a week after the results of the referendum, are we seeing an actual expression of thoughts of suicide? Or are these expressions a way of expressing the profundity of disappointment, anger or sadness at the outcome? This is as much as anything a cultural question I raise to the authors.

The inevitable extension of problematizing a sexual minority identity is expressed in lines 106-108 (“The result of such a study may provide empirical evidence to develop prevention and intervention strategies…”), and it is here that my concerns are most fully realised. This language effectively blames the victims for their own oppression. While yes, it is important that mental health strategies be in place for gender and sexually diverse persons, one would hope that these would not be limited to times of voter referendums, or that the solution to oppression of these groups lies in helping these groups accommodate to that oppression: the solution to oppression is to address the oppression, not to help people better accommodate to the oppression. The authors themselves allude to this point in lines 267-68: Whether civil rights of any individual should be determined through referendums should be “comprehensively evaluated”. I strongly encourage the authors to revise the abstract to include this latter sentence, since it really lies at the heart of this study.

Secondly, while the authors note that “access to Facebook is not yet [!!] universal” (line 322) they do not describe the penetration of Facebook in different age groups and sexualities across Taiwan. In other words, to what extent can the sample be considered ‘representative’ of Taiwan in general? The authors’ implication is that it is not, yet that needs to be stated clearly. If it is not representative, then what are the implications for the conclusions that the authors draw? The authors have undertaken an interesting and creative analysis on the data, but in all these analyses the data must meet basic statistical assumptions. If they do not then all the conclusions must be interpreted with great caution. For example, if young ‘out’ lesbians are more likely to be in isolated or remote communities, and therefore more likely to participate in Facebook as a way of creating a social community, are they more likely to express suicidal ideation? Sample bias must be considered in the analysis. I concur with the authors’ observation that longer follow up periods are required.

I am surprised that there is only one reference to Australian literature in the paper (and that related to relationships only). Australia is the only other ‘natural’ national experiment where a referendum (or ‘plebiscite’) on same-sex marriage was carried out (although the outcome was different as, of course is the culture).  The authors will want to consider Perales, Todd, et al. (2018), Perales, Reeves et al. (2018), Verrelli et al. 2019 and others, who considered how sexual minority communities in Australia were affected by homophobic publicity surrounding the plebiscite.  In fact the literature review is lighter on contemporary literature than one would expect for such a study: one-third (12/36) of the references are from the most recent five years (2015-2019), one third (12/36) from the last ten years (2010-2014) and one third (12/36) are older than ten years (≤2019). I would expect most references on this contemporary issue to be from the most recent five years.

Minor issues

Line 41 (and throughout): While I acknowledge that the taxonomy of sexuality and gender is fraught and contended, the authors apply ‘LGB’ language without explanation and somewhat uncritically. The authors are to be commended for recognising the spectrum of possible sexualities (asexual, pansexual, etc.), but then use LGB throughout. This acronym is western, and implies a static, categorical identity. May I invite the authors either to justify its use, or to consider more inclusive and less categorical language that more accurately reflects what in fact they did in their study?

Line 42 (“had almost two times and more than three times higher risks”) is unclear and must be reworded.

Line 85: my eyebrows raised that the use of ‘glorious’. I think the authors intend this to be ironic (which I applaud), but I must ask if this belongs in a scientific paper.

Author Response

Reviewer 3

Comment

Firstly, while I believe the authors of the paper are well-intended, same-sex sexuality and marriage equality is presented as a problem in the paper. This is to some extent inevitable, since the purpose of the paper is to consider a problem, but there is little recognition of the resiliency of gender and sexually diverse communities in Taiwan. To what extent is the expression of suicidal ideation a proxy for expressions of sadness, anger, or even disappointment? I am not suggesting that expressions of suicidal intent not be taken seriously. However, in English, for example, the expression “I was so embarrassed I could have died!” is not an expression of a wish to die, but rather an expression of the extent of embarrassment. Since the Wave 2 study was carried out only a week after the results of the referendum, are we seeing an actual expression of thoughts of suicide? Or are these expressions a way of expressing the profundity of disappointment, anger or sadness at the outcome? This is as much as anything a cultural question I raise to the authors.

Response

We appreciate the reviewer’s valuable comment. In the study we inquired participants’ suicidal ideation during the past week using a single question “Do you have any suicide ideation?” on the Revised 5-item Brief Symptom Rating Scale (BSRS-R). Just like the original purpose of developing the BSRS-5, this question was used for screening those with suicidal ideation and transferring them to mental health professionals for further help. A “yes” response to the question did not show the contexts that tormented the responders. Therefore, we listed it as one of the limitations of this study:

Fourth, participants’ suicidal ideation might develop in various biological, cognitive and emotional contexts. The present study did not clarify the mechanisms for the increased suicidal ideation rate in nonheterosexual participants. The results of previous studies may provide possible explanations for the mechanisms through which voter referendums affect the mental health of LGB individuals in the United States (US) [20,29,40]. However, whether these mechanisms proposed based on the US sociocultural background can well explain the increased suicidal ideation rate in nonheterosexual participants in Taiwan warrants further study.” Please refer to line 358-364.

Moreover, the Wave 2 study was carried out only a week after the results of the referendum, and the nonheterosexual participants might be in the state of anger and disappointment. We listed it as one of the limitations of this study. “Third, the Wave 2 survey was conducted 1 week after the The nonheterosexual participants might be in the state of huge anger and disappointment. Studies with relatively long follow-up periods are needed to examine the longer-term change in suicidal ideation.” Please refer to line 355-357.

Comment

The inevitable extension of problematizing a sexual minority identity is expressed in lines 106-108 (“The result of such a study may provide empirical evidence to develop prevention and intervention strategies…”), and it is here that my concerns are most fully realised. This language effectively blames the victims for their own oppression. While yes, it is important that mental health strategies be in place for gender and sexually diverse persons, one would hope that these would not be limited to times of voter referendums, or that the solution to oppression of these groups lies in helping these groups accommodate to that oppression: the solution to oppression is to address the oppression, not to help people better accommodate to the oppression. The authors themselves allude to this point in lines 267-68: Whether civil rights of any individual should be determined through referendums should be “comprehensively evaluated”. I strongly encourage the authors to revise the abstract to include this latter sentence, since it really lies at the heart of this study.

Response

Thank you for your comments. We revised the former sentence in the revised manuscript as below. “The result of such a study may provide empirical evidence to understand the impacts of voter-initiated referendums on mental health in minority groups whose rights are restricted or rejected, as well as to inspect whether civil rights of any individual can be determined through referendums.” Please refer to line 113-116. We also added the latter sentence into Abstract: Whether civil rights of sexual minority individuals can be determined through referendums should be evaluated.” Please refer to line 39-40.

Comment

Secondly, while the authors note that “access to Facebook is not yet [!!] universal” (line 322) they do not describe the penetration of Facebook in different age groups and sexualities across Taiwan. In other words, to what extent can the sample be considered ‘representative’ of Taiwan in general? The authors’ implication is that it is not, yet that needs to be stated clearly. If it is not representative, then what are the implications for the conclusions that the authors draw? The authors have undertaken an interesting and creative analysis on the data, but in all these analyses the data must meet basic statistical assumptions. If they do not then all the conclusions must be interpreted with great caution. For example, if young ‘out’ lesbians are more likely to be in isolated or remote communities, and therefore more likely to participate in Facebook as a way of creating a social community, are they more likely to express suicidal ideation? Sample bias must be considered in the analysis. I concur with the authors’ observation that longer follow up periods are required.

Response

Thank you for your comments. We revised the description regarding the limitation of recruiting participants in the present study as below: “First, although recruiting participants through Facebook can deliver large numbers of participants quickly, cheaply, and with minimal effort compared with mail and phone recruitment, access to Facebook is not universal to people of all age. A 2018 analysis found that 68.4% of active Facebook users in Taiwan were aged between 18 and 44 [38]. Moreover, people are not equally motivated to use Facebook [39]. Although the female: male ratio of Facebook uses in Taiwan is about 1:1 [38], no data shows the distribution of sexual orientation in Facebook users in Taiwan. Therefore, whether young lesbians are more likely to participate in Facebook as a way of creating a social community and more likely to express suicidal ideation warrants further study.” Please refer to line 343-351.

Comment

I am surprised that there is only one reference to Australian literature in the paper (and that related to relationships only). Australia is the only other ‘natural’ national experiment where a referendum (or ‘plebiscite’) on same-sex marriage was carried out (although the outcome was different as, of course is the culture).  The authors will want to consider Perales, Todd, et al. (2018), Perales, Reeves et al. (2018), Verrelli et al. 2019 and others, who considered how sexual minority communities in Australia were affected by homophobic publicity surrounding the plebiscite.  In fact the literature review is lighter on contemporary literature than one would expect for such a study: one-third (12/36) of the references are from the most recent five years (2015-2019), one third (12/36) from the last ten years (2010-2014) and one third (12/36) are older than ten years (≤2019). I would expect most references on this contemporary issue to be from the most recent five years.

Response

Thank you for your reminding. We reviewed the literature and found several valuable studies on the same-sex marriage referendum in Australia. We added them into the revised manuscript to improve the comprehensiveness of the literature review.

Line 258-260: Research in Australia found that more frequent exposure to negative media messages about same-sex marriage was associated with greater psychological distress [28].” Line 274-276: A previous study had a similar result that LGB people report comparatively worse life satisfaction, mental health and overall health in constituencies with higher rates voters saying “no” to same-sex plebiscite [33].” Line 299-300: Moreover, a longitudinal study in Australia found that nonheterosexual women were more disadvantaged in health and wellbeing than nonheterosexual men [37]. Line 375-377: “Research found that perceiving greater immediate social network can buffer the effect of exposure to negative media messages about same-sex marriage on psychological distress.” [28]. Line 377-378: “Perceived poor social support also mediates a large portion of the effects of structural stigma on LGB outcomes [33].”

Comment

Line 41 (and throughout): While I acknowledge that the taxonomy of sexuality and gender is fraught and contended, the authors apply ‘LGB’ language without explanation and somewhat uncritically. The authors are to be commended for recognising the spectrum of possible sexualities (asexual, pansexual, etc.), but then use LGB throughout. This acronym is western, and implies a static, categorical identity. May I invite the authors either to justify its use, or to consider more inclusive and less categorical language that more accurately reflects what in fact they did in their study?

Response

Thank you for your suggestion. In the revised manuscript we replaced “LGB” by “sexual minority” in most of contents and kept using “LGB” if the cited studies focused on LGB population. Please to throughout the manuscript.

Comment

Line 42 (“had almost two times and more than three times higher risks”) is unclear and must be reworded.

Response

We revised the sentence into “A meta-analysis found that sexual minority youth reported significantly higher rates of suicidality than did their heterosexual counterparts [1].” Please refer to line 46-48.

Comment

Line 85: my eyebrows raised that the use of ‘glorious’. I think the authors intend this to be ironic (which I applaud), but I must ask if this belongs in a scientific paper.

Response

Thank you for your reminding. We deleted the word “glorious” from the revised manuscript. Please refer to line 91.